# Insufficient antibody validation challenges oestrogen receptor beta research

Sandra Andersson[1], Mårten Sundberg[2], Nusa Pristovsek[1], Ahmed Ibrahim[3,4], Philip Jonsson[5,†], Borbala Katona[1], Carl-Magnus Clausson[1], Agata Zieba[1], Margareta Ramström[2], Ola Söderberg[6], Cecilia Williams[3,5,7] & Anna Asplund[1]

The discovery of oestrogen receptor β (ERβ/*ESR2*) was a landmark discovery. Its reported expression and homology with breast cancer pharmacological target ERα (*ESR1*) raised hopes for improved endocrine therapies. After 20 years of intense research, this has not materialized. We here perform a rigorous validation of 13 anti-ERβ antibodies, using well-characterized controls and a panel of validation methods. We conclude that only one antibody, the rarely used monoclonal PPZ0506, specifically targets ERβ in immunohistochemistry. Applying this antibody for protein expression profiling in 44 normal and 21 malignant human tissues, we detect ERβ protein in testis, ovary, lymphoid cells, granulosa cell tumours, and a subset of malignant melanoma and thyroid cancers. We do not find evidence of expression in normal or cancerous human breast. This expression pattern aligns well with RNA-seq data, but contradicts a multitude of studies. Our study highlights how inadequately validated antibodies can lead an exciting field astray.

[1] Department of Immunology, Genetics and Pathology, Uppsala University, Science for Life Laboratory, 751 85 Uppsala, Sweden. [2] Department of Chemistry, Uppsala University, Science for Life Laboratory, 75123 Uppsala, Sweden. [3] Division of Proteomics and Nanotechnology, School of Biotechnology, Science for Life Laboratory, KTH Royal Institute of Technology, 171 21 Solna, Sweden. [4] Division of Pharmaceutical Industries, National Research Centre, Dokki 12622, Egypt. [5] Department of Biology and Biochemistry, University of Houston, Houston, Texas 77204, USA. [6] Department of Pharmaceutical Biosciences, Uppsala University, 75124 Uppsala, Sweden. [7] Department of Biosciences and Nutrition, Karolinska Institutet, 141 83 Stockholm, Sweden. † Present address: Department of Epidemiology and Biostatistics and Human Oncology and Pathogenesis Program, Memorial Sloan Kettering Cancer Center, New York, New York 10065, USA. Correspondence and requests for materials should be addressed to C.W. (email: cecilia.williams@scilifelab.se).

O estrogen is a hormone with multiple roles in health and physiology. Importantly, it drives breast cancer (BC) growth and its inhibition is one of the most efficacious BC treatments to date. Oestrogen signalling is mediated by two oestrogen receptors that both belong to the nuclear receptor superfamily: ERα (*ESR1*) and ERβ (*ESR2*). ERα was identified in 1986 (refs 1,2), and became the first biomarker applied in oncology. Approximately 70% of all BCs overexpress ERα, and these tumours are targeted with selective oestrogen receptor modulators (SERMs, such as tamoxifen or raloxifene) or compounds that reduce endogenous oestrogen production (aromatase inhibitors). Immunohistochemical analysis (IHC) with a well-validated and specific antibody (1D5) is routinely used for treatment-predictive ERα analysis in clinical pathology[3,4]. Approximately 40% of ERα-positive BCs fail to respond or develop resistance to endocrine treatment[5]. Thus, upon the discovery of a second oestrogen receptor, ERβ, in 1996 (ref. 6), this was met with a massive interest for its potential to constitute a complementary treatment-predictive BC biomarker and therapeutic target. Multiple studies have further suggested ERβ as a plausible target for endocrine treatment of various other diseases, including benign prostate hyperplasia, prostate cancer and lung cancer[7,8].

However, after 20 years of intense studies, the role of ERβ and even its distribution of tissue and cellular expression are still unclear and debated. For example, variable ERβ expression in BC has been described in numerous publications[9–20], with contradicting correlations to clinical parameters (reviewed in refs 21–23). In addition, most cell lines have been reported to lack ERβ mRNA[22,24] (see also 45 cell lines analysed within the HPA project, http://www.proteinatlas.org/ENSG00000140009-ESR2/cell), while antibody-based applications report its protein expression[25,26]. This raises the pertinent question of antibody specificity.

Despite the widespread use of IHC, no universal scheme has been established to ascertain the functionality of an antibody before its use. This is increasingly recognized as a factor contributing to poor reproducibility of biomedical studies[27,28]. Standard means of validating antibody specificity, such as using pre-absorption with blocking peptide and/or western blot (WB) are recognized as crude assessments: blocking peptide does not control for unspecific binding in absence of the target protein, and a band in western blotting could correspond to many different proteins of approximately the same molecular weight. In addition, a comparative study of WB and IHC has shown that the performance of antibodies is application-dependent, and suggests that each antibody should be validated for the application it is intended for[29]. This is now reflected in recent guidelines for application-dependent validation of antibodies presented by the *ad hoc* International Working Group for Antibody Validation[30].

Efforts have been made to validate ERβ antibodies[31–33], but as a clear discrepancy between detectable mRNA and protein levels remains, and broadly accepted antibodies against ERβ generate disconcordant expression patterns, these efforts appear to have been insufficient. Our study aims to shed light on the controversies in this field by performing in-depth exploration of the antibodies' specificities and by defining accurate expression of ERβ. We use well-validated negative and positive controls and apply multiple antibody-based applications, including identification of bound protein by immunoprecipitation (IP) followed by mass spectroscopy (MS), at a scale that has not been previously undertaken. We demonstrate that only one of 13 antibodies is sufficiently specific in IHC. Applying this antibody, PPG5/10, for protein expression profiling of 44 normal and 21 tumour tissues within the Human Protein Atlas project, we detect ERβ protein only in testis, ovary, placenta (weakly), lymphoid cells, granulosa cell tumours, and a subset of malignant melanoma and thyroid cancers. We do not find evidence of expression in normal or cancerous human breast. This expression pattern aligns well with RNA expression data, but contradicts a multitude of studies. Our study highlights the importance of adequately validated antibodies.

## Results

**Most ERβ antibodies show false positivity.** To evaluate the specificity of ERβ-targeting antibodies, we screened 13 commercially available or in-house produced antibodies (Fig. 1; Supplementary Table 1), including the two most commonly used ones (monoclonals PPG5/10 and 14C8). Formalin-fixed and paraffin-embedded (FFPE) tissue specimens are the most prominent sample type at clinical pathology departments, and is thereby the format for which a clinically relevant antibody must be functional. Therefore, we first performed IHC on a validation tissue microarray (TMA) that included a panel of FFPE tissues and control cell lines (Supplementary Table 2). The cell line panel comprised four cell lines. The colon cancer HCT116 and BC T47D cell lines, which were confirmed to not express ERβ mRNA using RNA-seq (< 1 Fragments Per Kilobase and Million, FPKM, in this study), and qPCR[34]. HCT116 has also been shown to lack capacity to bind oestrogen according to competitive radioactive ligand-binding assay[35]. T47D is positive for ERα while HCT116 is not. Corresponding cell lines employing lentivirus-engineered expression of FLAG-tagged ERβ, were included as positive controls. The resultant expression and function of ERβ has been validated previously using multiple technologies[34–36]. Two mouse monoclonal antibodies (mAbs), PPZ0506 and 14C8, both displayed the expected nuclear staining in the two ERβ-expressing control cell lines, along with absence of positivity in the ERβ-negative control cell lines (Fig. 2a). However, 14C8 stained relatively few cells compared to the staining generated with PPZ0506. The remaining 11 anti-ERβ antibodies, including the widely used mAb PPG5/10 (Fig. 2a), all failed the IHC validation step since they generated distinct positive IHC staining in ERβ-negative cell lines.

A few selected tissues were also included in the validation TMA in order to control for representative staining quality (Supplementary Table 2). In these tissues, the PPZ0506 and 14C8 antibodies both stained a few cell nuclei in tonsil and Leydig cells in testis. Both also stained peripheral lymphocytes - PPZ0506 mainly in GI-tract and 14C8 in most normal tissues and cancers. Clone 14C8 displayed additional nuclear positivity in BC and colorectal cancer tissue which PPZ0506 did not. PPG5/10 showed a clear and widespread nuclear staining in tonsil, and in all Leydig cells and cells in seminiferous ducts in testis, as well as in non-malignant breast tissue, in BC, and in lymphocytes and glandular cells in GI-tract tissues. The staining patterns for both 14C8 and PPG5/10 correlated well with previously published reports using the respective antibodies. We could not find published studies applying PPZ0506 on clinical material, but observed that its staining pattern was notably more restricted than that of the other two antibodies.

For comparison and validation of protocols, the clinically approved ERα antibody (1D5) was analysed in the same manner. 1D5 displayed nuclear positivity in the ERα-positive cell line (T47D, Fig. 2a), along with expected staining in BC cases previously determined to be ERα-positive during clinical pathology assessment, in non-malignant breast tissue, and in a subset of cells in the germinal center of tonsil. No staining of the ERα-negative control cells (HCT116) was seen with this antibody.

Thus the ERα comparison demonstrated that the set-up and methodology performed as expected.

We conclude that out of the 13 ERβ-targeting antibodies only two, PPZ0506 and 14C8, appear to specifically target ERβ in FFPE-treated cell lines. These two, however, give a partially divergent staining pattern in tissues. The commonly used PPG5/10 was highly unspecific (Fig. 2a).

**Two antibodies recognize correct target in WB assay.** Although it is recognized that antibody performance is application dependent, WB remains the most commonly used assay for assuring specificity. To validate that the anti-ERβ antibodies 14C8 and PPZ0506 bind ERβ, and to further explore the performance of clone PPG5/10, these three antibodies were subjected to an extended antibody validation, including WB assay on positive and negative control cells (Fig. 2b; Supplementary Fig. 1). Recombinant ERβ protein (59 kDa) was included in a separate lane as reference. The PPZ0506 ERβ antibody displayed a single band of the expected size (60 kDa, the molecular weight of the expressed FLAG-tagged ERβ isoform 1) in lysates from positive control cells, and no distinct bands in the negative controls. Using clone 14C8, multiple bands including one of the correct size (59–60 kDa) were seen. This band was stronger in the positive control, but evident also in the negative control. However, storage for months of the 14C8 antibody rendered this antibody unable to recognize recombinant ERβ and subsequently the difference in intensity between the positive and negative control was no longer distinguishable (Supplementary Fig. 1b). The PPG5/10 antibody generated a strong unspecific band corresponding to 75–100 kDa, a molecular weight significantly larger than the ERβ protein in both positive and negative controls and did not show specificity for ERβ. A weak band similar to the correct size could occasionally be noted, in both positive and negative cells, along with a very weak detection of the recombinant ERβ (Fig. 2b) but this was usually not present (Supplementary Fig. 1c). WB using the ERα antibody clone 1D5 resulted in a single band corresponding to the expected size of ERα isoform 1 (66 kDa) in ERα-positive control cells only (Fig. 2b, right panel).

The results generated with the three ERβ antibodies using IHC and WB are not congruent. PPZ0506 and 14C8 both yielded the expected staining pattern using IHC on control cells, but 14C8 stained tissues that PPZ0506 did not and appears to also target additional proteins in WB. PPZ0506 did not generate false positivity in control cells in either IHC or WB, but did not stain tissues that are considered ERβ positive in the literature (most, however, based on previous analysis with antibodies 14C8 and PPG5/10). PPG5/10 displayed false positivity in IHC, and was also unspecific in the WB assay.

**Mass spectrometry analysis confirms affinity of PPZ0506.** In order to identify whether ERβ was indeed bound by these three antibodies, we performed IP followed by gel separation and MS. Sections corresponding to 50–80 kDa, encompassing the molecular weights of both ERα and ERβ, were analysed by MS in IP:s from a positive control cell line (ERα-negative HCT116 with transduced ERβ expression) in replicated experiments for each antibody. In concordance with the supportive IHC and WB results, the IP-MS analysis demonstrated, with a high level of confidence, that PPZ0506 binds ERβ (Fig. 2c; Supplementary Data 1). For the 14C8 and PPG5/10 antibodies, no significant ERβ hits were obtained when searching the human database. ERβ could however be detected at low confidence level in one of the two 14C8 replicates when adopting a directed search with an in-house constructed FASTA library (Supplementary Table 3). The

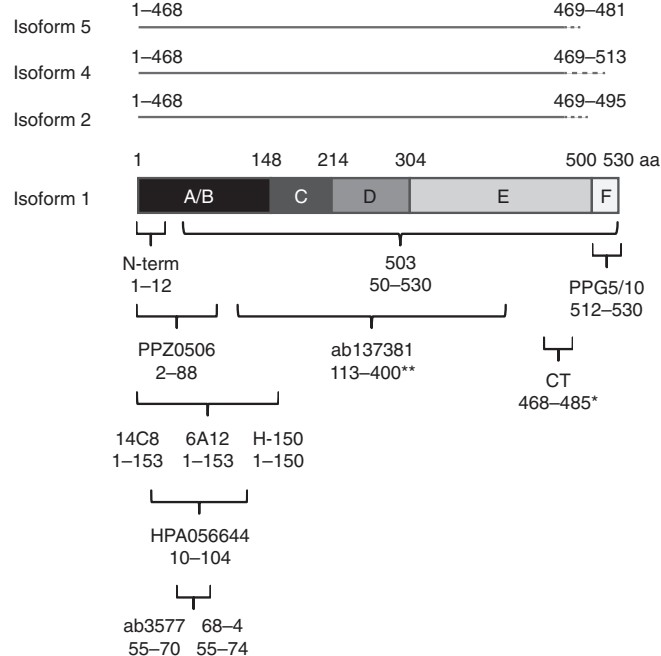

**Figure 1 | Schematic view of antibody epitopes and ERβ isoforms.** Overview of ERβ isoforms and regions targeted by ERβ antibodies included in the study. PPZ0506, 14C8 and PPG5/10 are indicated in bold. The structural domains of the ERβ protein (A,B: amino-terminal and activating function AF-1 domain, C: DNA-binding domain, D: hinge region (with nuclear localization and dimerization binding), E: ligand-binding domain and AF-2, and F: carboxyterminal end).

methodology control showed that ERα bound by antibody 1D5 could be detected by IP-MS, consistent with IHC and WB results (Fig. 2c; Supplementary Data 1).

We conclude that PPZ0506 robustly binds to ERβ protein, and that 14C8 may bind in a less reproducible fashion. We did not detect binding of ERβ by PPG5/10, and this correlates with our observation that this antibody did not clearly detect ERβ in neither WB nor IHC analysis.

**Unspecific bindings by 14C8 and PPG5/10 identified.** To identify unspecific bindings, peptides in the 50–80 kDa band as well as other bands visible on Coomassie brilliant blue-stained gel or WB were analysed by IP-MS, in both positive (HCT116 with transduced ERβ expression) and negative (ERα-positive T47D with no ERβ expression) control cells. Detailed results are provided as Supplementary Data 1. We found that ERβ protein was the dominating protein bound by PPZ0506. One more protein, WDCP, was detected in the MS analysis in both replicates of the positive control but not in the negative control, and only traces of other proteins were detected (Fig. 2c; Supplementary Data 1). However, ERβ was by far not the dominating protein in the IP-MS samples using the 14C8 antibody. Instead, a protein of the POU transcription factor family, POU2F1 (OCT1), was found with high significance in each replicate of both the positive and negative controls and in both the 50–80 kDa and 80–100 kDa gel bands. POU2F1 is a nuclear protein, with several isoforms of various sizes, of which one is nearly identical (58.7 kDa) to that of ERβ (59 kDa). The significant and robust hits for POU2F1 in the MS analyses strongly indicate that 14C8 binds this protein. As the ERβ transcript in this positive control cell line is expressed at four times higher level (FPKM = 25) than POU2F1 (FPKM = 6), 14C8 appears to preferentially bind POU2F1 over ERβ. For PPG5/10,

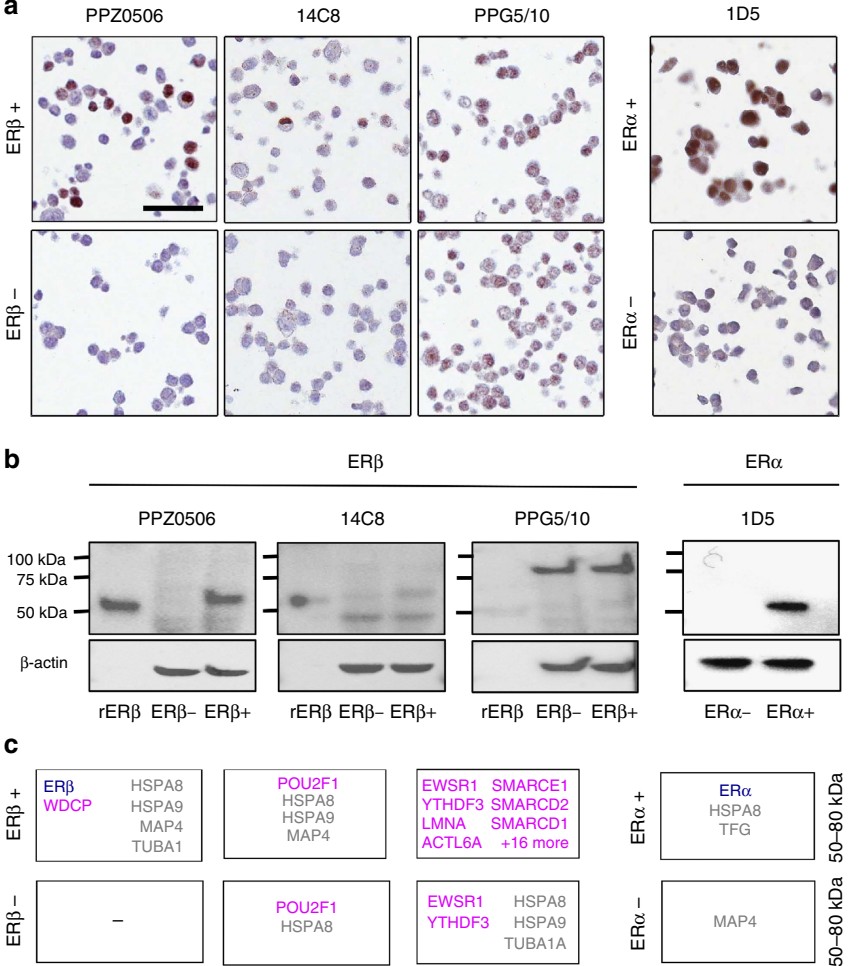

**Figure 2 | Validation of ERβ antibodies on positive and negative control cells.** (**a**) Representative images of IHC staining pattern on control cells. Left panel: ERβ antibodies PPZ0506 (Invitrogen, dilution 1:250), 14C8 (GeneTex, 1:1,500) and PPG5/10 (DAKO, 1:60) on ERβ-positive (HCT116-ERβ, top) and ERβ-negative (HCT116-Mock, bottom) control cells. Right panel: ERα antibody 1D5 (DAKO, 1:150) on ERα-positive (T47D-Mock, top) and ERα-negative (HCT116-Mock, bottom) control cells. Scale bar in top left image indicates 50 μm. (**b**) Representative images of western blotting on control cells. Left panels: Indicated ERβ antibodies (dilution 1:1,000, except for PPG5/10 which is 1:200) on recombinant ERβ (rERβ), ERβ-negative cell lysate (HCT116-Mock), and ERβ-positive cell lysate (HCT116-ERβ), performed on the same cell lysates run one gel on consecutive lanes. Right panel: ERα antibody 1D5 (1:1,000) on ERα-negative (HCT116-Mock) and ERα-positive (T47D-Mock) control cells. Lower panels show loading control (beta-actin). (**c**) Summary of proteins detected by MS in IPs of the respective antibodies in 50–80 kDa gel bands in replicated experiments. Blue font indicates expected antibody-targeted protein, purple non-intended targets, and grey proposed general binders.

the MS results indicate that this antibody targets a range of other proteins (Fig. 2c; Supplementary Data 1), most of which are nuclear according to gene ontology annotations. These include the transcriptional activator EWSR1 and YTHDF3 detected in both replicates of both positive and negative controls. We conclude that 14C8 appears to preferentially target POU2F1 over ERβ, and PPG5/10 targets multiple nuclear proteins but not ERβ.

**IHC with PPZ0506 is congruent with ERβ transcripts**. As part of a global expression profiling effort within the Human Protein Atlas (HPA) project, RNA-seq has been performed in a large panel of human normal and cancer tissues. These transcript levels may serve as blueprint for ERβ expression, guiding our effort to identify a specific antibody. We noted that the transcript data of normal tissues display a highly limited expression profile for ERβ: low expression levels in testis, adrenal gland, ovary, the GI-tract, and lymphoid organs (Fig. 3 and http://www.proteinatlas.org/ENSG00000140009-ESR2/tissue).

We applied IHC with antibodies PPZ0506, 14C8 and PPG5/10, respectively, on TMAs from the HPA encompassing 44 normal tissues, each represented by tissue samples from 3 different individuals (Supplementary Table 4).

With antibody PPZ0506, IHC positivity correlated well with the tissues' RNA expression and no staining was seen in tissues lacking detectable transcript levels, except for in placenta where a few weakly stained decidual nuclei could be observed. On the contrary, antibodies 14C8 and PPG5/10 resulted in distinct nuclear IHC positivity in a large number of tissues that did not display detectable levels of ERβ transcript. Figure 3a compiles the IHC positivity alongside the transcript values, and offers an overview of the congruency between transcript levels and PPZ0506 IHC staining, while exposing a disconcordant pattern when using antibodies 14C8 and PPG5/10. Corresponding data for ERα, where staining were only detected in tissues demonstrating high levels of mRNA, can be seen in Supplementary Fig. 2.

In Figure 3, representative IHC images, generated with each of the three ERβ antibodies, are shown for a subset of tissues. The upper panel shows that all three antibodies stain tissues that

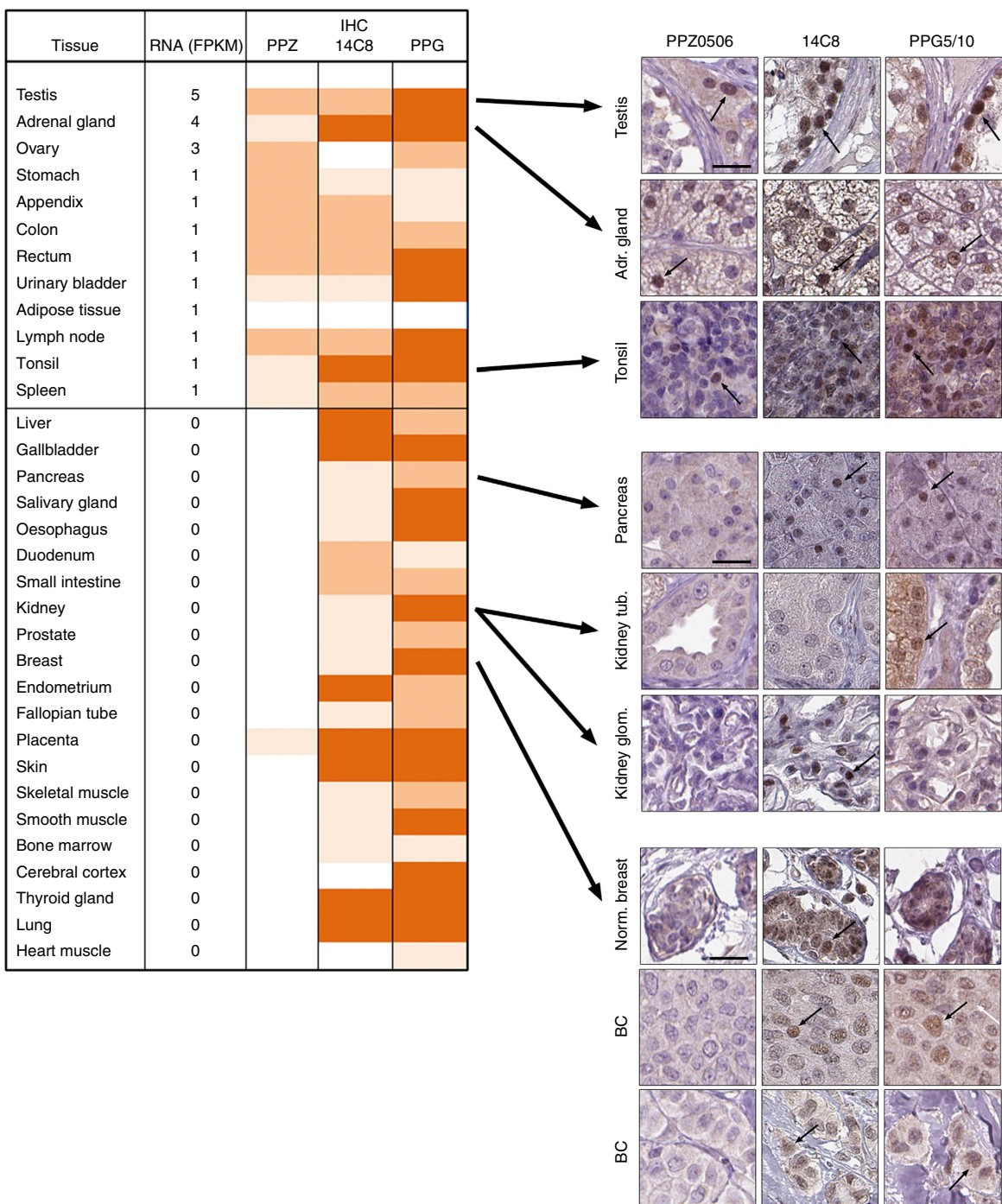

**Figure 3 | IHC with PPZ0506 but not 14C8 or PPG5/10 shows congruency with transcript levels in large panel of tissues.** Left panel: ERβ mRNA and protein expression in the Human Protein Atlas tissue panel. Average ERβ transcript levels (FPKM) measured using RNA-seq in triplicate tissue samples of 33 normal tissues displayed alongside annotated IHC positivity generated with antibodies PPZ0506 (1:600), 14C8 (1:1,500), and PPG5/10 (1:60), respectively. Annotated IHC positivity is dichotomized into grades 1-3, represented by colours (1: light brown, 2: medium brown, 3: dark brown). Right panel: Representative examples of IHC stainings generated using antibodies PPZ0506, 14C8 and PPG5/10, respectively. Top panel: Tissues with congruent IHC staining; Middle panel: tissues in which PPZ0506 generates a negative result while positivity is seen using 14C8 and PPG5/10; Lower panel: breast and breast cancer tissue in which positivity is seen using 14C8 and/or PPG5/10, while PPZ0506 generates a negative result. Brown: positive IHC staining; Blue: hematoxylin counter staining. BC: breast cancer. Scale bar in top left images indicates 25 μm. Arrows indicate examples of cells with nuclear staining.

express higher levels of ERβ mRNA. The middle and lower panels display tissues for which contradicting results are obtained with 14C8 and PPG5/10 compared to PPZ0506, and where ERβ transcripts are not detected.

Transcript data from the HPA and corresponding staining pattern of PPZ0506 are further supported by expression data from the Genotype-Tissue Expression (GTEx) consortium data[37]. The GTEx data comprise 53 normal tissues from multiple individuals and display ERβ transcript expression levels above 1.0 RPKM only in testis, adrenal gland, ovary and lymphocytes (Supplementary Fig. 3): This is highly concordant with RNA expression in HPA and staining pattern using PPZ0506.

Normal tissues

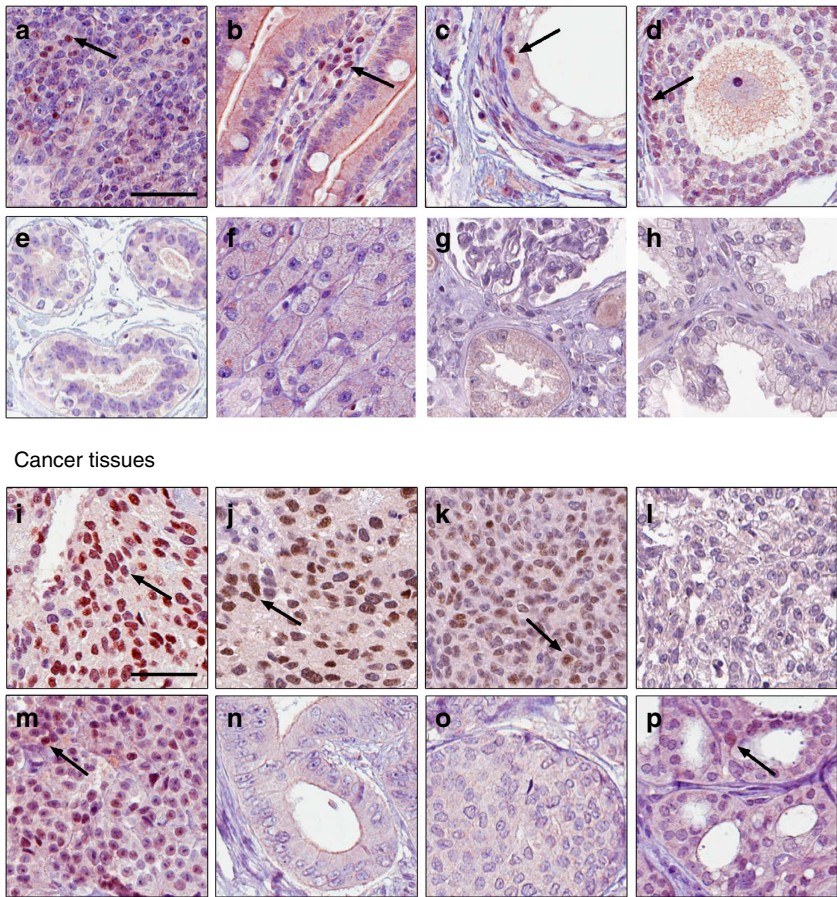

Cancer tissues

**Figure 4 | ERβ tissue profiling using antibody PPZ0506.** IHC profiling of normal tissues (top panel) and cancer tissues (lower panel). Nuclear ERβ positivity using PPZ0506 (dilution 1:600) is identified in normal tissues of (**a**) tonsil, (**b**) peripheral inflammatory cells (small intestine), (**c**) testis and (**d**) ovary. No IHC positivity was seen in (**e**) breast, (**f**) liver, (**g**) kidney, or (**h**) prostate. Nuclear ERβ positivity is identified in cancerous tissues (**i–l**) three out of four granulosa cell tumours, (**m**) one case of melanoma, and (**p**) one case of thyroid cancer. The remaining 16 cancer types were all IHC-negative, here exemplified by (**n**) colorectal cancer and (**o**) breast cancer. Brown: positive IHC staining; Blue: hematoxylin counter staining. BC: breast cancer. Scale bar indicates 50 μm. Arrows indicate examples of cells with nuclear staining.

In summary, we present evidence that when using IHC on FFPE tissues and cells, only PPZ0506 generate positivity supported by transcript levels.

**ERβ is expressed in endocrine and lymphoid tissues.** IHC serves as the golden standard of spatially resolved protein detection in tissues. After identifying PPZ0506 as a specific antibody for ERβ, we set out to map the expression pattern of ERβ in normal tissues and in a panel of common types of cancer. In addition to the 44 normal tissues ($n = 3$) above, we also analysed 21 malignant tumour types, ($n = 4$–12, Supplementary Table 4). As illustrated in Figs 3 and 4, and summarized in Table 1, IHC revealed nuclear ERβ expression in glandular cells of adrenal gland, granulosa cells in ovary, Leydig cells of the testis, and lymphocytes in secondary lymphoid organs, including tonsil, lymph nodes, and spleen, as well as in peripheral lymphocytes primarily in the intestinal tract. Most cancers did not display ERβ staining. However, in four out of five granulosa cell tumour cases, pronounced and distinct nuclear positivity was displayed in a substantial subset of tumour cells (Fig. 4i–k). Further, low to moderate expression was seen in a small subset of cells in 2 out of 12 cases of melanoma (Fig. 4m), and in 1 out of 4 cases of thyroid cancer (Fig. 4p). Positivity was also found in stromal cells in selected cases.

Altogether, ERβ was detected in 6 cell types in the 44 analysed normal tissues, and in 3 out of 21 cancer types examined (Figs 3 and 4; Table 1), with an emphasis on granulosa cell tumours, reproductive organs (testis and ovary), lymphocytes and lymphoid organs.

**ERβ expression is not detected in breast tissues.** As described above, IHC with clone PPZ0506 did not detect ERβ expression in normal breast or BC. This contradicts multiple studies, most of which have used antibodies 14C8 and PPG5/10 in IHC, and reported widespread ERβ expression in normal breast and BC (data compiled in Fig. 5a). No IHC studies concluding ERβ expression in breast cohorts have used the PPZ0506 antibody, the only truly specific antibody in our analysis. It is possible that the reported expression, or part of it, is due to unspecific staining. In order to look further into this discrepancy we evaluated whether compiled data on transcript levels of ERβ in a large number of BC samples might support IHC-positivity in some subgroups. A meta-analysis of RNA-seq data from 995 BC tumours and 99 normal breast tissues from The Cancer Genome Atlas (TCGA)[38] is compiled in Fig. 5b. We have also included levels for the related ERα, for which functionally consequences at the protein level can be deduced: Tumours that stain for ERα in routine clinical pathology assessment (using well-validated IHC

**Table 1 | Summary of ERβ protein expression in tissues.**

| Cell types | Tissues | Level of positivity for a certain cell type |
|---|---|---|
| Lymphocytes | Tonsil | Moderate to strong staining in few cells |
| | Lymph node | |
| | Spleen | |
| | Small intestine | |
| | Appendix | |
| | Colon | |
| | Rectum | |
| | Esophagus | |
| Granulosa cells | Ovary | Moderate staining in a subset of cells |
| Decidua cells | Placenta | Weak to moderate staining in few cells |
| Leydig cells | Testis | Weak to moderate staining in few cells |
| Stromal cells | Selected normal tissues | Weak to moderate staining in few cells |
| | Endometrial cancer | Weak staining in subset of endothelial cells in 1/12 individuals |
| Tumour cells | Granulosa cell tumour | Weak to strong staining of most cells in 4/5 individuals |
| | Melanoma | Weak to moderate staining of few cells in 2/12 individuals |
| | Thyroid cancer | Weak staining of few cells in 1/4 individuals |

All human cell types and tissues that show nuclear positivity using the antibody PPZ0506 in IHC are listed. These tissues, except placenta, also exhibit ERβ mRNA levels ($>1$ FPKM per RNA-seq analysis) in the Human Protein Atlas collection of 44 normal and 21 cancer tissue types.

protocols, denoted 'ERα +' in Fig. 5b) respond to endocrine treatment, whereas ERα-negative tumours do not. From this data we can estimate that an average ERα mRNA levels of $6\text{-}9_{\log2[\text{RSEMx}10^6+1]}$ resulted in detectable levels of protein, and that levels $<4.0_{\log2[\text{RSEMx}10^6+1]}$ typically did not generate detectable protein expression (Fig. 5b). As visualized in Fig. 5b, ERβ transcripts levels were less than $3.0_{\log2[\text{RSEMx}10^6+1]}$ in all normal and tumour samples, with the majority being $<1.0_{\log2[\text{RSEMx}10^6+1]}$. It is evident that transcript levels of ERβ are substantially lower, and that *none* of the 995 breast tumours, 99 normal breast specimens, or 7 distant metastases exhibit ERβ transcript levels expected to result in protein levels of detectable or functional consequences. Further, data from 214 normal breast tissues included in the GTEx data, show an average expression of ERβ transcripts less than 0.5 RPKM (Supplementary Fig. 3). Thus, we noted no subgroup of patients that had distinguishably higher levels, which could have indicated protein expression. We conclude that the lack of ERβ protein detection that we observe in normal breast and BC using IHC with PPZ0506 is supported by low to absent mRNA levels.

## Discussion

A lack of standardized guidelines for determining the specificity and functionality of antibodies has caused great discrepancies, lack of reproducibility, generation of dubious data, and significant amounts of wasted resources[27,28]. We here highlight the field of ERβ as an example of this problem. The inability to stringently detect this protein, as identified in this study, is likely one of the reasons for the lack of therapeutic success. As demonstrated in our study, the use of unspecific antibodies for ERβ detection remains a major obstacle in the field and, as a consequence, it is difficult to postulate and study its functional role and impact.

Only one antibody, PPZ0506, was demonstrated to target ERβ specifically using different affinity-based applications and controls (Figs 2 and 3; Supplementary Data 1). The other antibodies stained ERβ-negative cells and tissues and gave unspecific bands in WB or in Coomassie-stained SDS-PAGE of IPs. We used a semi-GeLC-MS approach to identify potential binders in corresponding IPs. One additional protein bound by PPZ0506, WDCP, was present only together with ERβ. While this could be an unspecific binding, we note that WDCP binds kinases and has been reported to interact with the SRC homology domain[39], and that the kinase SRC is a known ERβ-binding co-activator[40]. Thus, it is not inconceivable that the IP-MS detection of WDCP here indicates a true protein-protein interaction rather than unspecific binding. While PPZ0506 has been less used, 14C8 is the most referenced antibody for ERβ in Antibodypedia (http://www.antibodypedia.com/gene/11874/ESR2) and several previous reports find this antibody to work well[41–46]. We, however, noted that not only did 14C8 stain tissues that lack ERβ when applied in IHC, it also generated a band of the expected size in WB of negative controls that could easily be mistaken for ERβ. IP followed by MS indicated a greater affinity for the transcription factor POU2F1, which was consistently detected in the band ranges 50–80 and 100–120 kDa. POU2F1 is widely expressed, including in BC (Supplementary Fig. 4), and is implicated as a prognostic factor in prostate and gastric cancer[47,48]. Our study shows that applying this antibody with the aim of identifying ERβ can be misleading (Figs 2 and 3). The PPG5/10 antibody was found to target multiple nuclear proteins, including RNA-binding proteins (EWSR1 and YTHDF3), SWI/SNF family-member ARID1A, and several SWI/SNF-related matrix-associated actin-dependent regulators of chromatin subfamilies (SMARC proteins). Our findings can explain the clear nuclear staining these two antibodies generate in several tissues in absence of corresponding ERβ mRNAs. In Supplementary Data 1 we provide extended information of proposed specific, unspecific, and general binders (for example, MAP4, HSPA8, HSPA9), as generated by the IP-MS analysis for PPZ0506, 14C8, PPG5/10 and ERα antibody 1D5, and also highlight common MS contaminants[49]. As we did not analyse the ranges were no bands were visible in the IP-MS analysis, some less abundant binders may not have been detected.

Our results illustrate one example where the absolute majority of antibodies directed towards a protein were unspecific in IHC (12 out of 13; 92%). Considering the many studies that are performed on tissues that, according to results presented here, do not express the protein, this is an important negative finding.

Some of the antibodies evaluated in this study have been validated to various extends by the vendors. Recombinant protein is occasionally used (Upstate for Clone 68-4) and some show WB of cells transfected with ERβ (such as GeneTex for 14C8). Although such validations show that the antibody can detect its intended target when it is overexpressed, it does not demonstrate specificity under endogenous conditions. Other companies use

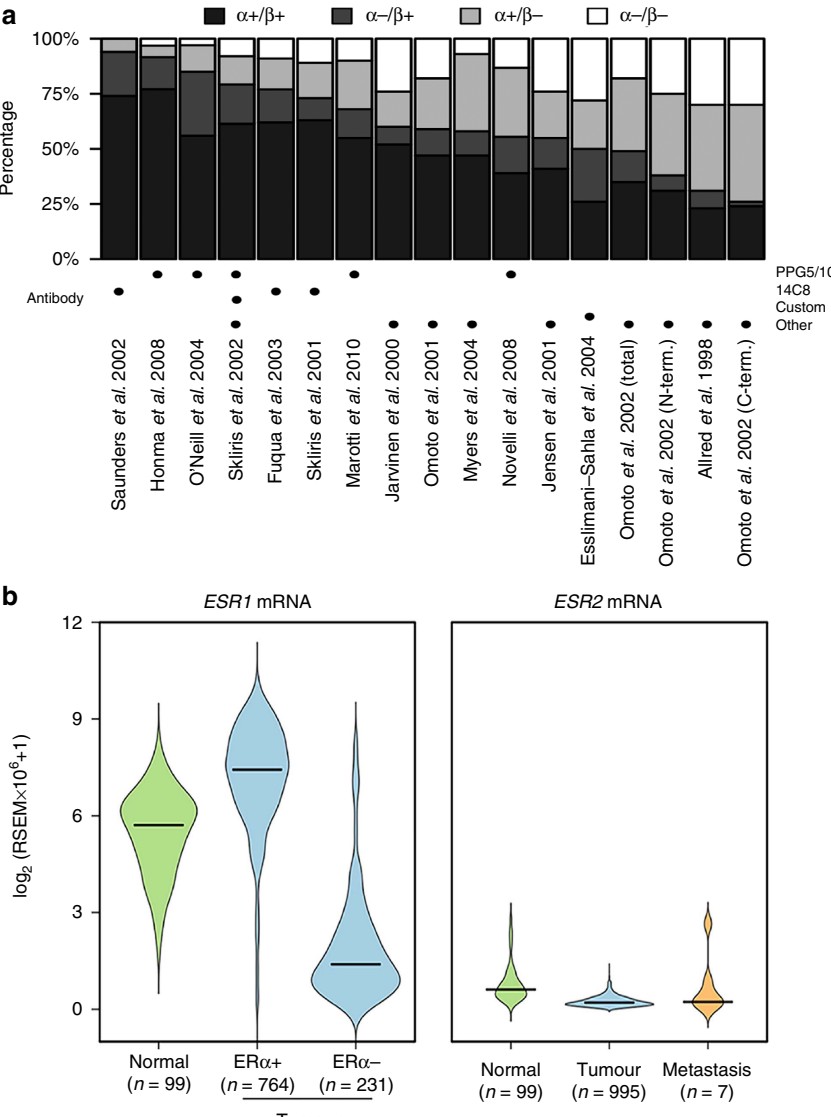

**Figure 5 | Compilation of ERβ expression data in breast.** (**a**) ERα and ERβ protein expression determined using IHC in 17 different studies. The total percentages of ERα- and/or ERβ-positive BC cases are shown. The ERβ antibody used in each study is indicated below. (**b**) RSEM expression estimates of ERα (ESR1) and ERβ (ESR2) mRNA expression in TCGA's BRCA cohort, including 99 normal breast specimens, 995 breast tumours, and 7 distant metastases. For the ERα-expression graph (left), the tumour samples are divided into ERα-positive and ERα-negative status based on clinical pathology diagnosis. For ERβ (right panel) samples are divided into normal, tumour and metastatic samples. The violin plots show probability densities for the expression in the indicated subsets of the cohort, with the medians indicated by black crossbars.

WB of cell lines that do not express ERβ (such as HeLa, MCF-7, Hek293), but still demonstrate clear strong bands of expected size (e.g. Abcam for Ab137381 and Ab133467, Upstate for Clone 68-4). While for example PPG5/10 has been reported to not work well in the WB application, it has still been considered specific in IHC applications[32], and several vendors refer predominantly to supporting IHC results (e.g. Biorad PPG5/10, DAKO, Thermofisher, Novus, Abcam). As illustrated in this study, these validations appear insufficient.

Our results corroborates previous studies that, based on RT-PCR, found the highest mRNA expression of ERβ in ovary, testis[50] and granulosa cell tumours[51]. In high concordance with our results, studies have also described ERβ protein expression in granulosa cells of the ovary and Leydig cells of the testis[52]. The latter study performed IHC using a rabbit pAb (06–629, Upstate Biotechnology), which thus appears to have generated results in accordance with PPZ0506. However, the same study

also described staining of the prostate, something we could not detect and which was not supported by transcript levels (Figs 3 and 4; Supplementary Fig. 3). The Upstate antibody was directed against a synthetic peptide corresponding to amino acids 46–63 of the N terminus of human ERβ. For comparison, the PPZ0506 was generated towards amino acids 2–88. As the Upstate antibody was a polyclonal, it was a limited recourse and the same batch is no longer available. Current antibodies from Upstate did not score well in our analysis. As implicated by its expression in reproductive organs, an effect of ERβ on human fertility is plausible. Corrresponding effect has also been demonstrated in knockout animals, where females are sub-fertile[53–55]. Impact on fertility is one of few functional ERβ effects that are widely accepted and reproducible (reviewed in ref. 56). Further, our data presented here support studies of ERβ as a potential biomarker in granulosa cell tumours, and in a subset of melanoma and thryroid tumours.

As ERβ was originally cloned from a rat prostate cDNA library[6], its expression and functions in the prostate have been studied extensively. IHC, using the PPG5/10 and other antibodies, have determined ERβ expression in both normal prostate and in prostate cancer[57], and ERβ-positivity has been correlated to primary Gleason grade and Gleason score in prostate cancer[26]. However, its role in prostate cancer remains unclear (reviewed in ref. 58), and our analysis with the PPZ0506 antibody did not corroborate significant expression in the human prostate. ERβ mRNA did not reach detectable levels in the three individuals included in the HPA expression profiling using RNA-seq (Fig. 3, left panel). GTEx data show mRNA expression in normal prostate to be similar to levels in bladder and spleen (approx. 0.5 RPKM, Supplementary Fig. 3), although higher in some individuals. It is possible that there are species-related differences in the expression of ERβ or that analysis of larger cohorts may determine expression in a minority of cases.

Although multiple studies report expression of ERβ in BC, our conclusion is that the antibodies used to detect abundant expression of ERβ in breast and other tissues are not sufficiently specific. Similar IHC-based studies have also described ERβ protein expression in numerous other tissues, including lung and brain (reviewed in ref. 8), although expression of ERβ transcript appears to be very limited or absent (Human protein atlas: http://www.proteinatlas.org/ENSG00000140009-ESR2/tissue; GTEX: http://www.gtexportal.org/home/gene/ESR2; BioGPS: http://biogps.org/#goto=genereport&id=2100). IHC with PPZ0506 in our study did not detect protein expression in these tissues.

Finally, several isoforms of ERβ has been annotated. Isoform 2 (ERβcx), 4, and 5 all differ from isoform 1 by their C-terminal region, but are otherwise identical (Fig. 1). The PPZ0506 and 14C8 antibodies are both directed towards the N-terminal region, and would thus target all isoforms. PPG5/10, on the other hand, is one of few antibodies directed solely towards isoform 1, as its epitope is in the C-terminal end of the ERβ protein (Fig. 1). This is one reason this antibody is favoured in many studies. However, our positive controls specifically express isoform 1 and the poor specificity of this antibody when applied in IHC, as demonstrated here, is deeply concerning.

We conclude that the PPZ0506 constitutes a specific but not frequently used ERβ antibody. Applying this antibody, we demonstrate that the ERβ protein is expressed in a limited number of normal and cancer tissue types, with the highest expression detected in granulosa cell tumours. Expression in most human tissues, including in breast and prostate, was undetectable. Although the tissue staining pattern generated with the antibodies 14C8 and PPG5/10 in our study correlated well to previous reports, we could demonstrate that these antibodies stain multiple tissues that lack detectable transcript levels. Our results imply that numerous published studies with broadly accepted anti-ERβ antibodies have described ERβ expression incorrectly. While our study focuses on ERβ, we do not think that antibodies towards ERβ are significantly poorer than those targeting other proteins, and it is not unlikely that this problem generates similar obstacles in many other fields. Our study illustrates the consequences of using antibodies not specific for the intended targets. We wish to argue for the need of a more thorough validation from the vendors, and a higher level of consciousness among buyers, about the fact that each antibody must be validated for its intended application using proper controls.

## Methods

**Cell lines and culture conditions.** Four control cell lines with defined expression of ERα and/or ERβ (Supplementary Table 2) were used for evaluation of the panel of antibodies (Supplementary Table 1). The ductal BC cell line T47D expresses ERα but not ERβ, while colon cancer cell line HCT116 does not express either of the ER-receptors. As no cell line exist with a significant expression of ERβ (ref. 24), we used cell lines (originally acquired from ATCC, Rockville, MD) which has previously been transduced to express full length ERβ, and with corresponding controls (Mock)[36]. The resulting ERβ mRNA has been sequenced, and the protein has been shown to be expressed, bind ligand, and have functional effects[34–36,59]. We used these four control cell lines: T47D-ERβ (ERα+/ERβ+), T47D-Mock (ERα+/ERβ−), HCT116-ERβ (ERα−/ERβ+) and HCT116-Mock (ERα−/ERβ−) as positive and negative controls. T47D cells were cultured in 50% DMEM low glucose (SIGMA, ref D6046) and 50% F-12 (SIGMA, ref N6658) and HCT116 cells in RPMI-1640 (SIGMA, ref R0883). Both media were supplemented with 5% FBS (Gibco, ref 10270), 1% PEST (SIGMA, ref P0781), as well as Blasticidin (5 µg ml$^{-1}$; Invitrogen, ref R210-01) to maintain ERβ expression in transduced cells.

**Harvesting and protein extraction.** Subconfluent (70%) cells were harvested using a Non-Interfering Protein Assay Kit (Calbiochem, ref 488250) and protein extraction was performed using either ProteoExtract Complete Mammalian Proteome Extraction Kit (Calbiochem, ref 53977) according to the manufacturers' instructions, or RIPA-buffer (Sigma, ref R0278) with 1% Protease Inhibitor Cocktail (Sigma, ref P8340) and 0.15% Benzonase (Calbiochem, ref KP31255). For RIPA-buffer, frozen cell pellets were resuspended in the supplemented RIPA-buffer and incubated for 20 min at 4 °C followed by centrifugation at 21,000 rcf at 4 °C. Protein extract supernatants were stored at −70 °C.

**Tissue microarray.** Human tissue samples used for protein and mRNA expression analyses were collected and handled as routinely performed within HPA. In short, tissues were obtained from the Department of Pathology, Uppsala University Hospital, Uppsala, Sweden as part of the sample collection governed by the Uppsala Biobank (http://www.uppsalabiobank.uu.se/en/). All human tissue samples used in the present study were de-identified in accordance with Swedish laws and regulations, and approval and advisory from the Uppsala Ethical Review Board (Reference # 2002–577, 2005–338, 2007–159, and 2011–473). FFPE tissue samples were collected from the pathology archives based on hematoxylin-eosin (HE) stained tissue sections showing representative normal histology for each tissue type. Representative cores (1 mm diameter) were sampled from the FFPE blocks and assembled into TMAs. Control cell lines were also included and processed through formalin-fixed paraffin-embedded procedures[60], and aims to mimic standard FFPE-treatment of tissue[61]. The validation TMA contained 8 tissue types as well as FFPE-processed control cell lines (Supplementary Table 2), and the profiling TMAs ($n=9$) included 44 normal tissue types and 21 different cancer types (4–12 patient samples/cancer; Supplementary Table 4). The TMA blocks were cut in 4-µm sections and placed on Superfrost Plus microscope slides (Thermo Fisher Scientific; Fremont, CA), dried in RT overnight and baked in 50 °C for 12–24 h before IHC.

**Antibodies.** Thirteen ERβ antibodies: PPZ0506 Invitrogen (Catalogue number 417100, manufactured by Perseus proteomics PPMX); 14C8 Gene Tex (GTX70174); PPG5/10 DAKO (M7292), BioRad (MCA1974G1), and Thermofisher (MA1-81281); 6A12 Novus (NB200-303); ab133467 Abcam (ab133467); 68-4 Upstate (05-824); ERb_503 (in-house); H150 Santa Cruz (sc-8974); N-terminal (in-house); ab137381 Abcam (ab137381); CT Upstate (07-359); HPA056644 Human Protein Atlas (in-house); ab3577 Abcam (ab3577); and one ERα antibody (1D5, DAKO, M7047) were examined. Information about each antibody, including lot numbers, is provided in Supplementary Table 1, and their epitopes are illustrated in Fig. 1.

**Immunohistochemistry.** The TMA slides were deparaffinized in xylene, hydrated in graded alcohols, and blocked for endogenous peroxidase for 5 min in 0.3% $H_2O_2$ diluted in 95% ethanol. Heat-induced epitope retrieval (HIER) was performed in a decloacing chamber (Biocare Medical; Walnut Creek, CA) at 121 °C for 10 min in citrate buffer pH 6.0 (Thermo Fisher Scientific, ref. TA-250-PM1X). The slides were rinsed in distilled water, and immersed in wash buffer (Thermo Fisher Scientific, ref. TA-999-TT) containing 0.2% Tween 20 (Thermo Fisher Scientific, ref. TA-125-TW) for 15 min to eliminate surface tension. The staining was performed at room temperature in an automated instrument, Autostainer 480 S (Thermo Fisher Scientific), and the slides washed with wash buffer between all steps. The slides were incubated with UltraV block (Thermo Fisher Scientific, ref. TA-125-UB) for 5 min, followed by primary antibody for 30 min (optimized for dilutions between 1:50 and 1:1500, final dilutions used as indicated in Supplementary Table 1 and in corresponding figure legends), primary antibody enhancer (Thermo Fisher Scientific, ref. TL-125-PB) for 20 min, UltraVision LP HRP polymer (Thermo Fisher Scientific, ref. TL-125-PH) for 30 min, and finally diaminobenzidine (DAB; Thermo Fisher Scientific, ref. TA-125-HDX) for 10 min. The slides were counterstained with Mayer's hematoxylin (Histolab; Gothenburg, Sweden, ref. 01820), dehydrated, and coverslipped using Pertex (Histolab, ref. 00871.0500) in a Leica AutoStainer XL instrument.

**Annotation.** Tissues included in the TMAs were manually annotated based on observed nuclear positivity. Positive parenchymal cells and immune cells, fibroblast and endothelial cells in stroma were categorized as weak, moderate or strong. The estimated fraction of positive cells was described either as few (less than 25%), medium (around 25–75%) or most (more than 75%). In Fig. 3 and Supplementary Fig. 2, combinations of intensity and fraction were converted into a numeral score (1–3) for each annotated tissue according to: 1 – weak/few, weak/medium or moderate/few; 2 – weak/most, moderate/medium or strong/few; and 3 – moderate/most, strong/medium or strong/most.

**Western blot.** Whole protein was extracted from three cell lines (HCT116-ERβ, HCT116-Mock and T47D-Mock) using RIPA lysis buffer (Thermo Scientific) supplemented with protease and phosphatase inhibitor (Thermo Scientific) according to the manufacturer's protocol. The lysates were incubated on ice for 30 min, vortexed 10 min, and spun down at 13,000 r.p.m. at 4C for 20 min. Total protein concentration was determined using DC protein assay reagents (Bio-Rad), and included different BSA dilutions for standard curve. For each sample, 60 µg protein was loaded in 4-20% SDS-precast protein gel (Bio-Rad). Recombinant full length ERβ (530 amino acids, ~59 kDa) was used as additional positive control for ERβ detection and size determination. Protein ladder Precision Plus Protein Kaleidoscope Prestained Protein Standards (#161-0375, BioRad) was used. Separated proteins were then transferred to PVDF blot using Transblot Turbo transfer kit for 10 min and Transblot Turbo Transfer System (Bio-Rad). Transfer efficiency was checked by immersing the blot in Ponceau S red staining (Fluka), after which, the blot was destained by washing with TBST buffer. The blot was blocked in 5% non-fat dry milk (Bio-Rad) for 1 h at room temperature, washed three times with TBST, 5 min each. Blots were incubated with different primary antibodies against ERβ: PPZ0506 (1:500 and 1:1,000), 14C8 (1:300, 1:500 and 1:1,000), PPG5/10 (1:200, 1:500 and 1:800) and for ERα: ID5 (1:1,000). B-actin (Santa Cruz # SC-47778) was used as loading control. All primary antibodies were mouse IgG isotypes and diluted in 1% NFDM-TBST. Anti-mouse IgG HRP-conjugated secondary antibody (Cell Signaling) was used and incubated with the blot in 5% NFDM-TBST for 1 h at room temperature. To visualize the specific bands, Clarity western ECL substrate (Bio-Rad) was used and the images were captured by ChemiDoc MP imaging system (Bio-Rad). WB was repeated at least four times for each ERβ antibody.

**Immunoprecipitation.** Two µg of each antibody were incubated at +4 °C for 4 h with the protein extract. The antibody/protein-mix was further incubated at +4 °C for 1 h with Dynabeads Protein G from the IP Kit (ref. 100.07D, Life technologies, Carlsbad, CA). The beads were washed three times with washing buffer, and the antibodies removed from the beads with elution buffer according to the manufacturer's recommendations. The samples were stored at −70 °C. Two IP replicates for each ERβ antibody were performed at different occasions.

**SDS-PAGE and mass spectrometry.** SDS-PAGE, in-gel tryptic digestion and LC–MS/MS analysis were performed as described by Sundberg et al.[62] with some adjustments. In short, the IP samples were suspended in a volume of 5 µl 5xLaemmli buffer with an addition of 2.5% 2-mercaptoethanol and were loaded on an 18-well TGX Criterion precast gel, 4-20% (BioRad Laboratories). The electrophoresis was run at 200 V for 45 min in Tris/glycine/SDS running buffer and gels were stained with Coomassie blue R-250 (BioRad Laboratories), according to manufacturer's instructions. Gel bands corresponding to 50–80 kDa, including the molecular weights of ERα and ERβ, and additional bands observed in WB or SDS-PAGE (20–25, 80–120 and 120–135 kDa) were collected and divided into smaller pieces (~1 mm³). Gel pieces were destained by washing (25 mM NH₄HCO₃ and 100% CAN) twice or until sufficient colour had been removed, and dried (SpeedVac system for 15 min). 10 mM DTT was added and incubated at 50 °C for 1 h, followed by 1 h incubation in 50 mM IAA at room temperature in darkness. Then, washing and drying procedure was repeated. A solution of 12.5 ng ml⁻¹ trypsin was added and tryptic digestion performed overnight in darkness at 37 °C. The solution was transferred to a new test tube and peptides were extracted from gel slices in a solution of 60% ACN and 5% FA during sonication for 5 min. That solution was pooled with the previous fraction and the samples were completely dried in SpeedVac. The peptides were re-suspended in 20 µl of 0.1% FA and analysed on a nanoLC-LTQ-Orbitrap Velos Pro ETD mass spectrometer. An EASY-nLCII system (ThermoFisherScientific) was used for the on-line Nano-LC separation. 5 µl of the sample was loaded onto a pre-column (EASY-Column, C18-A1, ThermoFisher Scientific) at a maximum pressure of 280 bar. The peptides were then eluted onto to an EASY-column (C18-A2, ThermoFisher Scientific), used for the separation. The separation was performed at a flow rate of 200 nl min⁻¹ using mobile phase A (0.1% FA) and B (99.9% CAN, 0.1% FA). A 2-step 90 min gradient, 2% B up 50% B in 75 min followed by wash step of 100% B for 15 min was applied. The mass spectrometer was equipped with a nano-flex ion source. The spray voltage was set to 2.0 kV. The instrument was controlled through Tune2.6.0 and Xcalibur 2.1 and operated in data dependent mode to automatically switch between high-resolution mass spectrum and low resolution in the LTQ. The survey scan was performed from m/z 400– 2,000 at 100,000 resolution and the 10 most abundant ion peaks were CID fragmented for

each full scan cycle. The mass window for precursor ion selection was set to 1.9 Th. Screening was done for charge state +2, +3 and +4 and the dynamic exclusion was set to 30 s. Normalized collision energy of 35%, activation time of 10 ms and activation q of 0.25 were set for MS/MS. The fragments were scanned at 'normal scan rate' in the low-pressure cell of the ion trap and detected with a secondary electron multiplier. Two replicated runs were performed with each of the three ERβ antibodies, and one with the ERα antibody.

**Data analysis.** For protein identification, Proteome Discoverer, version 1.4.1.14 (ThermoFisher Scientific) was used. A gene ontology (GO) (www.geneontolo-gy.org) annotation was done with the support of the Proteome Discoverer software. The searches were performed against the Uniprot human FASTA library downloaded from www.uniprot.org (2013-12-02) applying Sequest HT in combination with Percolator. Sequest HT searches were also conducted against a smaller in-house constructed FASTA library containing only 29 oestrogen receptor-related proteins (Supplementary Table 3) and for those searches, no Percolator was used. These additional analyses were performed to detect possible trace amounts of the target proteins. The parameters for the search were set to: fixed modifications: carbamidomethyl (C), variable modifications: deamidated (N, Q) and oxidation (M), precursor mass tolerance: 10 p.p.m., fragment mass tolerance: 0.6 Da and maximum two missed cleavage sites. The S/N threshold was set to 1.5. The search results were validated using the Percolator algorithm and an FDR of 5%. A minimum of two unique peptides per protein was applied.

For analysis of TCGA data: RNA-seq data consisting of 946 breast tumour and 107 normal breast samples were acquired from TCGA[38]. Expression values were normalized transcript count estimates, according to the RSEM algorithm for RNA-seq analysis[63]. Expression was illustrated in Fig. 5b as violin plots, where areas represent expression in the different BC subsets (luminal A, luminal B, basal-like, normal-like and Her2-enriched, as well as divided by normal tissue, primary tumour and metastasis), and lines across indicate median value.

**Data availability.** The MS data have been deposited to the ProteomeXchange Consortium via the PRIDE[64] partner repository (data set identifier [PXD005936]). Public RNA-seq data were accessed from the Human Protein Atlas (proteinatlas.org), the GTEx Portal (http://gtexportal.org) and TCGA (now available at the NCI's Genomic Data Commons http://gdc.cancer.gov). RNA-seq data for control cell lines are included in Supplementary Data 1 as FPKM values. The complete set of immunohistochemistry images for PPZ0506 over human normal and tumour tissues is scheduled for public release in the next version of the Human Protein Atlas (proteinatlas.org, planned for Oct 2017).

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

## Acknowledgements

We would like to thank Lars-Arne Haldosén (Karolinska Institutet), Jan-Åke Gustafsson, Margaret Warner, Christoforos Thomas and Anders Ström (all University of Houston) for insightful discussions regarding ERβ expression and antibodies, and Emma Lundberg (Royal Institute of Technology KTH) for discussions regarding antibody validation. Further, we thank Jun Wang (University of Houston) for laboratory assistance. The results shown here are in part based upon data generated by the TCGA Research Network: http://cancergenome.nih.gov/ and The Genotype-Tissue Expression (GTEx) Project: http://www.gtexportal.org/. This work was supported by grants from the National Cancer Institute at the National Institutes of Health (R01CA172437 to C.W.), Marie Curie Actions FP7-PEOPLE-2011-COFUND (GROWTH 291795) via the Swedish Governmental Agency for Innovation Systems (VINNOVA) programme Mobility for Growth (to C.W.), the Swedish Cancer Society (to C.W.), the Stockholm County Council (SLL, to C.W.), the Swedish Research Council (to O.S.), and the Knut and Alice Wallenberg foundation.

## Author contributions

S.A., C.-M.C., A.Z. and N.P. performed IHC experiment, A.I. and S.A. performed WB experiments; M.S. performed MS experiment, M.R. advised MS experiment, P.J. performed meta analyses, S.A., M.S. and B.K. analysed data; S.A., B.K., M.S., C.W. and A.A. interpreted results of experiments; S.A., A.I., P.J., B.K. and A.A. prepared figures; S.A. drafted manuscript; S.A., B.K., C.W. and A.A. edited and revised manuscript; all

authors approved final version of manuscript; O.S. advised on antibody specificity, C.W. initiated and designed the study, supervised P.J. and A.I., and advised on oestrogen receptors, and A.A. coordinated the study and supervised.

## Additional information

**Competing interests:** A.A. and M.S. are as HPA researchers stakeholders in Atlas Antibodies through the IP holding company Atlasab Intressenter. The remaining authors declare no competing financial interests.

**DOI: 10.1038/ncomms16164**      **OPEN**

# Corrigendum: Insufficient antibody validation challenges oestrogen receptor beta research

Sandra Andersson, Mårten Sundberg, Nusa Pristovsek, Ahmed Ibrahim, Philip Jonsson, Borbala Katona, Carl-Magnus Clausson, Agata Zieba, Margareta Ramström, Ola Söderberg, Cecilia Williams & Anna Asplund

*Nature Communications* 8:15840 doi: 10.1038/ncomms15840 (2017); Published 15 Jun 2017; Updated 29 Nov 2017

In this Article, two papers are mistakenly listed as having made use of the antibody 14C8 instead of the antibody PPG5/10. In the Discussion section, ref. 45 is incorrectly cited as having shown that the antibody 14C8 works well, and in Fig. 5a, Saunders *et al.* 2000 is incorrectly depicted as using the 14C8 antibody. Both these papers used antibody PPG5/10 and neither paper includes experiments using 14C8. A corrected version of Fig. 5 appears below as Fig. 1.

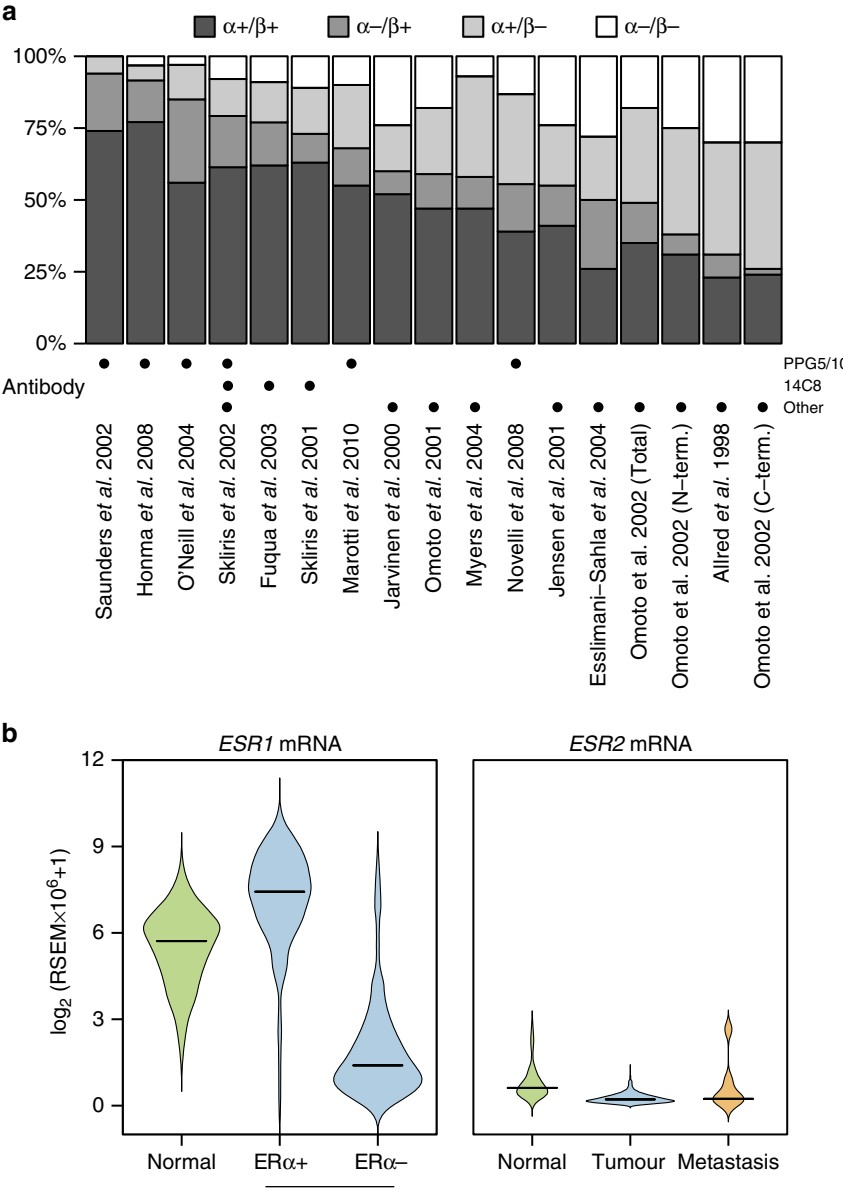

**Figure 1**

