## [Peer Review File · Nature Communications]

Reviewers' comments:

Reviewer #1, an expert in mass spectrometry (Remarks to the Author):

This paper presents detailed validation results for a series of antibodies, several of which are routinely used for clinical analysis, to measure estrogen receptor beta (ERb). The results are cautionary tale as most of the Abs tested lack specificity or recognize other proteins with greater specificity. These Abs have been used in large numbers of published studies resulting in conflicting and likely erroneous results in many of these reports, and more importantly possibly preventing advances that could have been made in translational research for detection and treatment of a range of cancers. The authors go on to identify one, rarely used mAb that does work well and then used this Ab to measure ERb levels in a range of normal and tumor tissues, demonstrating that the expression patterns observed using this Ab now align well with RNA-seq expression data (as they should). Perhaps most disturbing of all is the finding using this Ab that ERb is not expressed at detectable levels in either normal or cancerous human breast tissue, contradicting the results of numerous studies.

This is an important and well conducted study, with proper controls and appropriate orthogonal methods especially affinity-mass spectrometry for direct identification of binders. The study builds on an increasingly sad story of how inadequate validation of Abs can lead to decades of misspent research efforts and lost opportunities for useful translational research.

There are a number of points, most easily addressable, that need to be dealt with. However, the MS results need to be substantially clarified and re-reviewed as the validity of these findings cannot be assessed due the absence of key results and data. This concern and others more easily addressed are detailed, below.

1. The presentation of the results of the affinity-mass spectrometry is inadequate and somewhat confusing, which is disappointing given the overall focus on validation and that this is one of the best, if not the best method for establishing what was bound and detected by these Abs.

Specifically, Table 1 only presents the results for the 50-80kDa region of the gels where ERb should be found by AP-MS using three of the Abs. Only ERalpha and beta are identified by name, while a column labeled "Proteins Identified" only shows the number of proteins (in addition to Era or ERb??) that were detected in that band without identifying what they are (gene name, protein name, what their expected MWs are, numbers and the sequences of peptides presented, etc.) as is standard practice. In contrast, Supplemental Table 4 presents gene names and protein names as well as GO annotation, for proteins observed using the single Ab PPG5/10, but not the other two presented in Table 1. However there is no way to correlate in MW region each protein was identified, or what its expected MW is, or other information required to be presented noted above. As a result, neither Table 1 or Suppl. Table 4 are of much use in their current forms. If authors want to keep Table 1 in maintext (this reviewer is not convinced it should be there), then at least the other proteins identified in each band and their expected MWs should be shown as well as the number of peptides identified for each protein identified (not just sequence coverage). This Table could then be supplemented by an appropriately expanded version of Suppl. Table 4, preferably in a useful and searchable format such as Excel rather than pdf, that lists proteins, sequences of peptides id'ed etc. for ALL of the Abs evaluated. Alternatively, one unified table should be created that has all of the key information noted above and the table placed in Supplement.

2. Related to the above, all raw data, together with a document describing clearly what is in each file, must be deposited in a public repository and the link provided in the paper.

3. This reviewer is disappointed that current standard practice of GelC MS was not followed, i.e., where the entire lane is cut into bands and analyzed, not just what is visibly stained by Coomassie. There were also no controls run (i.e., IgG on bead or similar) to try to tweeze out generic off-target protein binders vs. off-target proteins specifically enriched by the Abs, with the exception of the 100 kDa band. While this does not affect the conclusions as to what was bound in the 50-80kDa or 100 kDa band regions, it was a lost opportunity to compare relative and absolute protein

abundances (as determined by MS, with its caveats) for all proteins enriched by each Ab.

4. Line 165, page 8: the sentence describes that PPG5/10 detects a 100 kDa band, but figure 1 shows that it is 14C8 that detects such a band. Either the figure labeling or this sentence appear to be in error.

5. Discussion is somewhat redundant with the Results section and should be shortened.

6. Supplementary Figure 5 is quite valuable. This reviewer would replace Table 1 in maintext with this figure.

7. Related to this, more information should be provided in Discussion as to how/if the vendors validated their Abs.

8. Lot numbers for the Abs used MUST be provided.

9. The recent paper by Uhlen and colleagues should be cited (Uhlen Nature Methods 2016). This paper presents a range of methods used for Ab validation, detailing their capabilities and limitations, together with a proposal for how validation of antibodies should be done going forward in the field.

Reviewer #2, an expert in antibody validation (Remarks to the Author):

MAJOR CLAIMS:

The authors show they were able to successfully perform a rigorous validation of 13 different antibodies to ER-beta, and show convincingly that the majority of antibodies (commercially available or produce in-house) are non-specific, including the two antibodies most widely used in research studies and pathology labs (PPG5/10 from Dako and 14C8 from Gene Tex). The authors use IP paired with mass spectrometry to confirm what proteins the non-specific antibodies are binding to, and explain how they could have been confused for ER-beta in previous studies (nuclear expression, similar size). They conclude that only one antibody is specific and passes all components of validation (PPZ0506 from Invitrogen).

The authors furthermore show that specificity is dependent on context of use. Some antibodies appear specific by western blot, but are not specific using tissue IHC, and vice versa. Therefore, antibodies must be validated across multiple orthogonal platforms, and especially for the specific intended context of use.

In this manuscript, the authors set the standard for how rigorous and thorough antibody validation should be. This includes 1) IHC on a variety of known positive and negative cell line controls (known negative cell lines plus matched cell lines expressing FLAG-tagged protein of interest); 2) western blot (WB) performed on similar panel of cell lines; 3) immunoprecipitation (IP) paired with mass spectrometry to confirm specificity of antibody and identify any non-specific proteins an antibody binds.

Use of the appropriately-validated antibody on human tissues for the first time (44 normal and 21 cancerous) shows where ER-beta is actually expressed, which includes only a few normal tissues (testis, ovary, placenta, and lymphoid cells) and some cancers (most granulosa cell tumors, and a subset of thyroid cancers and melanoma). The authors show these findings correlate well with expression using RNA-seq. The evidence to date on ER-beta expression in normal and cancerous breast (which conflicts with the authors' findings) appears most likely due to the use of non-specific antibodies in prior studies. The use of non-specific antibodies in general has resulted in the plethora of contradictory data on the role of ER-beta.

The authors suggest these findings (large numbers of non-specific antibodies being widely used and resulting in erroneous published data) are not unique to ER-beta, but likely could be the cause of existing contradictory data for multiple potential biomarkers and therapeutic targets. If we hope to develop effective therapeutic targets and biomarkers in the future, antibodies should be validated according to standards that these authors demonstrate.

ARE THE FINDINGS NOVEL, AND WILL THEY BE OF INTEREST TO OTHERS IN THE FIELD?

The findings in this manuscript are definitely novel, and will be of interest (should be of interest!) to many in the clinical pathology field as well as many researchers. For ER-beta specifically, this paper shows convincingly that the two antibodies most frequently used are clearly binding non-specific nuclear proteins of similar size to ER-beta (they also show for the first time what non-specific proteins these are), and data using these antibodies can't be trusted. This has massive implications for interpretation of existing data in the literature, as well as how studies should be performed going forward. The authors show unambiguously, and for the first time, that only one commercially-available ER-beta antibody can be trusted (PPZ0506 from Invitrogen). This is the only antibody that should be used in future studies, and the quality of data in this manuscript should give scientists and clinicians confidence in using this antibody for ER-beta studies in the future.

The model for validating antibodies, while not entirely novel (see Bordeaux J et al, Antibody Validation, Biotechniques, 2010; PubMed link: <https://www.ncbi.nlm.nih.gov/pmc/articles/PMC3891910/>), is one which sets a rigorous and thorough standard, and should serve as a model for others going forward.

The findings should also serve as a sobering warning to others, and should encourage investigators and clinicians to utilize extreme caution in both interpreting existing data as well as utilizing antibodies (to any target) in the future. The clinical impact for the Estrogen Receptor and breast cancer fields is also very high. This is the first manuscript I'm aware of that raises clear and consistent doubts about the existing ER-beta data in breast cancer. I think these implications are critical to the field going forward and hopefully will receive adequate attention.

ANY FURTHER EVIDENCE REQUIRED?

I found the results to be very convincing and clear. I don't think any further experiments are required to strengthen the authors' conclusions. There was one very minor thing that might help to have clarification in the text. How were the negative control cell lines used (HCT116 and T47D) confirmed to lack ER-beta expression? I trust the data and think it's convincing as is, but would be nice to have extra description in the text here.

Overall, I think the detailed conclusions are well-stated and very convincing, and the final message of caution is one that I hope readers take very seriously.

WILL THE MANUSCRIPT INFLUENCE THINKING IN THE FIELD?

I believe this manuscript should definitely influence how the current literature on ER-beta is interpreted, and should provide investigators with tools they can be confident in going forward. I believe it should also influence the way we think about using antibodies, and the critical importance of proper validation. While there has been growing awareness about this in recent years (and the authors have noted this), I think more evidence is always important, especially such a focused 'case study' as this. While it provides specific insight into only one biomarker, I think this is how validation must be performed (one-by-one) before antibody-based studies on each target can be confidently performed and their data trusted.

I think the finding that ER-beta does not appear to be present in normal or cancerous breast tissue is an especially relevant one on its own, as there is extensive data and discussion of this currently in the literature. If this manuscript can have high visibility in the breast cancer and Estrogen Receptor field, I believe it will have huge implications on how current data on the role of ER-beta

in breast cancer is interpreted. In addition, the convincing data that ER-beta is, in fact, expressed significantly in granulosa cell tumors (as well as at lower levels in a subset of thyroid cancers and melanomas), provides useful insight into the potential biology of these tumor types, and definitely suggests the role of ER-beta as a biomarker should be further studied in these tumor types.

NOTES ON STATISTICAL ANALYSES AND LEVEL OF DETAIL

There were no incredibly complicated or sophisticated statistical analyses performed in this study, but I think the level and validity of methods used were appropriate for the needs and purpose of the study. The authors performed these experiments with a sufficient level of detail and reproducibility, utilizing multiple different normal and cancerous tissues, and multiple (often 2 or 3) different tissue cores for each. The use of IP with mass spec for ER-beta antibodies was essential, not only to prove the non-specificity of certain antibodies, but also to reveal what non-specific proteins they are binding to, and what the existing data may actually be showing.

Point-by-point response to the referees' comments

We are thankful for the insightful comments offered by the reviewers. Please see our response to each of the issues raised below. We have also edited the text throughout.

Reviewers' comments:

Reviewer #1, an expert in mass spectrometry (Remarks to the Author):

1. The presentation of the results of the affinity-mass spectrometry is inadequate and somewhat confusing, which is disappointing given the overall focus on validation and that this is one of the best, if not the best method for establishing what was bound and detected by these Abs.

Specifically, Table 1 only presents the results for the 50-80kDa region of the gels where ERb should be found by AP-MS using three of the Abs. Only ERalpha and beta are identified by name, while a column labeled "Proteins Identified" only shows the number of proteins (in addition to Era or ERb??) that were detected in that band without identifying what they are (gene name, protein name, what their expected MWs are, numbers and the sequences of peptides presented, etc.) as is standard practice. In contrast, Supplemental Table 4 presents gene names and protein names as well as GO annotation, for proteins observed using the single Ab PPG5/10, but not the other two presented in Table 1. However there is no way to correlate in MW region each protein was identified, or what its expected MW is, or other information required to be presented noted above. As a result, neither Table 1 or Suppl. Table 4 are of much use in their current forms. If authors want to keep Table 1 in maintext

(this reviewer is not convinced it should be there), then at least the other proteins identified in each band and their expected MWs should be shown as well as the number of peptides identified for each protein identified (not just sequence coverage). This Table could then be supplemented by an appropriately expanded version of Suppl. Table 4, preferably in a useful and searchable format such as Excel rather than pdf, that lists proteins, sequences of peptides id'ed etc. for ALL of the Abs evaluated. Alternatively, one unified table should be created that has all of the key information noted above and the table placed in Supplement.

Authors:

We appreciate this valuable feedback and have thoroughly re-worked our presentation of the mass spectrometry data. We now provide a complete supplementary table in excel as Supplementary Data 1. Information of the identified peptides and corresponding proteins, including gene name, protein name, expected MW, molecular function, and more is shown at Level 1. On Level 2 (i.e. in excel, click on '+' of corresponding row), information of all identified peptides is provided. We have consequently removed Table 1 from the manuscript, included Supplementary Data 1, and summarized the MS data in Fig. 1c. We have adjusted the text accordingly.

2. Related to the above, all raw data, together with a document describing clearly what is in each file, must be deposited in a public repository and the link provided in the paper.

Authors:

The mass spectrometry data and file description have been deposited to the ProteomeXchange Consortium via the PRIDE partner repository with the data set identifier PXD005936. This information is now included in Material and Methods under Data availability (p. 26).

3. This reviewer is disappointed that current standard practice of GeLC MS was not followed, i.e., where the entire lane is cut into bands and analyzed, not just what is visibly stained by Coomassie. There were also no controls run (i.e., IgG on bead or similar) to try to tweeze out generic off-target protein binders vs. off-target proteins specifically enriched by the Abs, with the exception of the 100 kDa band. While this does not affect the conclusions as to what was bound in the 50-80kDa or 100 kDa band regions, it was a lost opportunity to compare relative and absolute protein abundances (as determined by MS, with its caveats) for all proteins enriched by each Ab.

Authors:

We agree GeLC-MS would have provided a more complete picture of all binders. However, since the purposes of these experiments were to confirm the binding of the target protein and to correlate the results of what was observed by WB, we selected a semi-GeLC-MS-approach. By combining the information from WB and Coomassie-stained SDS-PAGE, the gel's regions of interest were selected and analyzed. For the antibody with poorest specificity, PPG5/10, we analyzed nearly the entire lane as four gel regions covering 20-25, 50-80, 80-110, and 120-135 kDa were analyzed. For the most specific antibody, PPZ0506, WB detected one strong band and solely the region around the expected molecular weight 50-80 kDa was analyzed. (The results of each gel band are now provided in Supplemental Data 1, see point 1 above).

While we did not analyze IgG, which we agree would have been an appropriate control, we did analyze IP with four different antibodies, and we regard them as internal controls for each other. To utilize the comparison between antibodies and to enhance reading of Supplemental Data 1, we have included several subheadings: *Targeted protein* which lists the protein the antibody was intended towards along with result of the MS analysis; *Proposed unspecific targets* lists the detected proteins we suggest are off-target binders specifically enriched by the antibody analyzed; *Proposed general binders* is our attempt to tweeze out generic off-target protein binders by identifying those bound in common by several different antibodies; *Skin & Saliva-related proteins (proposed contaminants)* lists detected peptides from proteins known or predicted to be of skin or saliva-origin; and *Contaminants – other* lists those other detected proteins which lack RNA expression in the samples analyzed, and therefore are likely contaminants (rather than binders). We include the corresponding RNA-expression data for all proteins detected, and for each sample. We interpret an RNA-seq value of $FPKM \leq 1$ as no expression (highlighted in table). Finally, we cross reference the published CRAPome (Mellacheruvu *et al.*, Nature Methods 2013) and indicate those proteins proposed to be common general binders/contaminants (with red font). We have expanded on the discussion of the MS results (Discussion section p. 15-16), and also include the rational and limitations of our approach in the discussion in the text.

4. Line 165, page 8: the sentence describes that PPG5/10 detects a 100 kDa band, but figure 1 shows that it is 14C8 that detects such a band. Either the figure labeling or this sentence appear to be in error.

Authors:

Both 14C8 and PPG5/10 were found to frequently bind larger size proteins (75-100 kDa) in WB. PPG5/10, as discussed in the manuscript, gave two very different blots: one specific (which was included in Fig. 1b) and one highly unspecific, where it generated large bands in positive and negative cells alike (this was mentioned, but not shown). However, while scrutinizing our data during the revision, we realized that the specific blots with PPG5/10 appeared highly similar to the ones generated using PPZ0506. We have now backtracked the experiments in detail, and have found that all blots with PPG5/10 that yielded the specific results were performed with antibody from the same aliquoted batch of antibody. Other batches (and lot numbers) *all* rendered the unspecific pattern. We believe that the PPZ0506 antibody was *mislabelled* or *misread* during aliquoting with the quite similar name PPG5/10 (written on a tube, this can look nearly identical), at one occasion. This presumed mislabeling affected our interpretation of the performance of PPG5/10 in western blot assays -- but it did not affect any other experiment, the immunohistochemistry or mass spec (which were not performed with the mislabeled aliquot). We have gone to great length confirming and correcting this, we have repeated the PPG5/10 WB with both old and new PPG5/10 antibody, and from several different vendors and lot numbers, and can conclude that no other PPG5/10 than the alleged mislabeled aliquot gives a specific blot. As the PPG5/10 antibody performed poorly in all assays, it was surprising to us that it would work well in western blotting, and we now show that it does, in fact, not. We have adjusted the text to reflect the correct analyses, and we have run a new gel for Fig. 1b where the same protein extracts have been run side by side and then blotted with PPZ0506, 14C8, and PPG5/10, respectively. We also provide four or five replicated full-sized blots for each of these antibodies, performed at both labs (Uppsala and Stockholm), by different personnel, at different time-points, and with antibodies from different batches, lot numbers and/or vendors, as Supplementary Information (Supplementary Figure 1).

5. Discussion is somewhat redundant with the Results section and should be shortened.

Authors: The redundant sections have been removed.

6. Supplementary Figure 5 is quite valuable. This reviewer would replace Table 1 in maintext with this figure.

Authors: We have followed this advice (former Supplementary Figure 5 is now replaced with new Fig. 1, and Table 1 removed).

7. Related to this, more information should be provided in Discussion as to how/if the vendors validated their Abs.

Authors: We now include this information in the discussion (p. 16-17).

8. Lot numbers for the Abs used MUST be provided.

Authors: We now include both catalogue number and lot numbers in Supplementary table

1.

9. The recent paper by Uhlen and colleagues should be cited (Uhlen Nature Methods 2016). This paper presents a range of methods used for Ab validation, detailing their capabilities and limitations, together with a proposal for how validation of antibodies should be done going forward in the field.

Authors: This study is now cited.

Reviewer #2, an expert in antibody validation (Remarks to the Author):

MAJOR CLAIMS:

The authors show they were able to successfully perform a rigorous validation of 13 different antibodies to ER-beta, and show convincingly that the majority of antibodies (commercially available or produce in-house) are non-specific, including the two antibodies most widely used in research studies and pathology labs (PPG5/10 from Dako and 14C8 from Gene Tex). The authors use IP paired with mass spectrometry to confirm what proteins the non-specific antibodies are binding to, and explain how they could have been confused for ER-beta in previous studies (nuclear expression, similar size). They conclude that only one antibody is specific and passes all components of validation (PPZ0506 from Invitrogen).

The authors furthermore show that specificity is dependent on context of use. Some antibodies appear specific by western blot, but are not specific using tissue IHC, and vice versa. Therefore, antibodies must be validated across multiple orthogonal platforms, and especially for the specific intended context of use.

In this manuscript, the authors set the standard for how rigorous and thorough antibody validation should be. This includes 1) IHC on a variety of known positive and negative cell line controls (known negative cell lines plus matched cell lines expressing FLAG-tagged protein of interest); 2) western blot (WB) performed on similar panel of cell lines; 3) immunoprecipitation (IP) paired with mass spectrometry to confirm specificity of antibody and identify any non-specific proteins an antibody binds.

Use of the appropriately-validated antibody on human tissues for the first time (44 normal and 21 cancerous) shows where ER-beta is actually expressed, which includes only a few normal tissues (testis, ovary, placenta, and lymphoid cells) and some cancers (most granulosa cell tumors, and a subset of thyroid cancers and melanoma). The authors show these findings correlate well with expression using RNA-seq. The evidence to date on ER-beta expression in normal and cancerous breast (which conflicts with the authors' findings) appears most likely due to the use of non-specific antibodies in prior studies. The use of non-specific antibodies in general has resulted in the plethora of contradictory data on the role of ER-beta.

The authors suggest these findings (large numbers of non-specific antibodies being widely used and resulting in erroneous published data) are not unique to ER-beta, but likely could be the cause of existing contradictory data for multiple potential biomarkers

and therapeutic targets. If we hope to develop effective therapeutic targets and biomarkers in the future, antibodies should be validated according to standards that these authors demonstrate.

ARE THE FINDINGS NOVEL, AND WILL THEY BE OF INTEREST TO OTHERS IN THE FIELD?

The findings in this manuscript are definitely novel, and will be of interest (should be of interest!) to many in the clinical pathology field as well as many researchers. For ER-beta specifically, this paper shows convincingly that the two antibodies most frequently used are clearly binding non-specific nuclear proteins of similar size to ER-beta (they also show for the first time what non-specific proteins these are), and data using these antibodies can't be trusted. This has massive implications for interpretation of existing data in the literature, as well as how studies should be performed going forward. The authors show unambiguously, and for the first time, that only one commercially-available ER-beta antibody can be trusted (PPZ0506 from Invitrogen). This is the only antibody that should be used in future studies, and the quality of data in this manuscript should give scientists and clinicians confidence in using this antibody for ER-beta studies in the future.

The model for validating antibodies, while not entirely novel (see Bordeaux J et al, Antibody Validation, Biotechniques, 2010; PubMed link: <https://www.ncbi.nlm.nih.gov/pmc/articles/PMC3891910/>), is one which sets a rigorous and thorough standard, and should serve as a model for others going forward.

The findings should also serve as a sobering warning to others, and should encourage investigators and clinicians to utilize extreme caution in both interpreting existing data as well as utilizing antibodies (to any target) in the future. The clinical impact for the Estrogen Receptor and breast cancer fields is also very high. This is the first manuscript I'm aware of that raises clear and consistent doubts about the existing ER-beta data in breast cancer. I think these implications are critical to the field going forward and hopefully will receive adequate attention.

ANY FURTHER EVIDENCE REQUIRED?

I found the results to be very convincing and clear. I don't think any further experiments are required to strengthen the authors' conclusions. There was one very minor things that might help to have clarification in the text. How were the negative control cell lines used (HCT116 and T47D) confirmed to lack ER-beta expression? I trust the data and think it's convincing as is, but would be nice to have extra description in the text here.

Authors: We now describe the negative controls in greater detail (p. 6, and p. 22) and reference studies where this has been investigated. We also include the mRNA levels of ERbeta of respective negative and positive control cell lines in Supplementary Data 1 (as FPKM values).

Overall, I think the detailed conclusions are well-stated and very convincing, and the final message of caution is one that I hope readers take very seriously.

WILL THE MANUSCRIPT INFLUENCE THINKING IN THE FIELD?

I believe this manuscript should definitely influence how the current literature on ER-beta is interpreted, and should provide investigators with tools they can be confident in going forward. I believe it should also influence the way we think about using antibodies, and the critical importance of proper validation. While there has been growing awareness about this in recent years (and the authors have noted this), I think more evidence is always important, especially such a focused 'case study' as this. While it provides specific insight into only one biomarker, I think this is how validation must be performed (one-by-one) before antibody-based studies on each target can be confidently performed and their data trusted.

I think the finding that ER-beta does not appear to be present in normal or cancerous breast tissue is an especially relevant one on its own, as there is extensive data and discussion of this currently in the literature. If this manuscript can have high visibility in the breast cancer and Estrogen Receptor field, I believe it will have huge implications on how current data on the role of ER-beta in breast cancer is interpreted. In addition, the convincing data that ER-beta is, in fact, expressed significantly in granulosa cell tumors (as well as at lower levels in a subset of thyroid cancers and melanomas), provides useful insight into the potential biology of these tumor types, and definitely suggests the role of ER-beta as a biomarker should be further studied in these tumor types.

NOTES ON STATISTICAL ANALYSES AND LEVEL OF DETAIL

There were no incredibly complicated or sophisticated statistical analyses performed in this study, but I think the level and validity of methods used were appropriate for the needs and purpose of the study. The authors performed these experiments with a sufficient level of detail and reproducibility, utilizing multiple different normal and cancerous tissues, and multiple (often 2 or 3) different tissue cores for each. The use of IP with mass spec for ER-beta antibodies was essential, not only to prove the non-specificity of certain antibodies, but also to reveal what non-specific proteins they are binding to, and what the existing data may actually be showing.

REVIEWERS' COMMENTS:

Reviewer #1 (Remarks to the Author):

All of my concerns have been adequately addressed.

Point-by-point response

REVIEWERS' COMMENTS:

Reviewer #1 (Remarks to the Author):

All of my concerns have been adequately addressed.

Authors: The reviewers had no further concerns, and we have responded to the editorial requests separately, in the cover letter.